# Daylight-Active Cellulose Nanocrystals Containing Anthraquinone Structures

**DOI:** 10.3390/ma13163547

**Published:** 2020-08-11

**Authors:** Yiwen Zhu, Audrey Sulkanen, Gang-Yu Liu, Gang Sun

**Affiliations:** 1Department of Biological and Agricultural Engineering, University of California, Davis, One Shields Avenue, Davis, CA 95616, USA; yiwzhu@ucdavis.edu; 2Department of Chemistry, University of California, Davis, One Shields Avenue, Davis, CA 95616, USA; asulkanen@ucdavis.edu (A.S.); gyliu@ucdavis.edu (G.-Y.L.)

**Keywords:** cellulose nanocrystal, anthraquinone, reactive oxygen species, FTIR, atomic force microscopy

## Abstract

Antimicrobial and antiviral materials have attracted significant interest in recent years due to increasing occurrences of nosocomial infections and pathogenic microbial contamination. One method to address this is the combination of photoactive compounds that can produce reactive oxygen species (ROS), such as hydrogen peroxide and hydroxyl radicals to disinfect microbes, with carrier materials that meet the application requirements. Using anthraquinone (AQ) and cellulose nanocrystals (CNCs) as the photoactive and carrier components, respectively, this work demonstrated the first covalent incorporation of AQ onto CNCs. The morphology and the photoactive properties were investigated, revealing the structural integrity of the CNCs and the high degree of photoactivity of the AQ-CNC materials upon UVA exposure. The AQ-CNCs also exhibited an unexpected persistent generation of ROS under darkness, which adds advantages for antimicrobial applications.

## 1. Introduction

Antimicrobial and antiviral materials have been a growing field of interest for researchers due to increasing nosocomial infections and pathogenic microbial contaminations. Considerable effort has been devoted to developing surfaces with biocidal functions and self-disinfecting properties [1,2,3], as such modified surfaces could be incorporated into sterile garments and fabrics, thereby reducing the possibility of microbial and/or viral contamination. 

In this pursuit, the functionalization of materials with light-reactive groups could be used to reduce microbial contamination and viral infection [4]. Species that can absorb long-wavelength ultraviolet light (UVA) or daylight and become excited to produce reactive oxygen species (ROS), including hydrogen peroxide, hydroxyl radicals, and even singlet oxygen [5,6], are ideal candidates as functional groups. The ROS would provide antibacterial and self-disinfecting properties for the modified materials, as well as being environmentally friendly since they only require light exposure and oxygen to function [7,8,9,10]. 

In selecting a light-reactive species, anthraquinone derivatives are promising candidates, as they are known for being light-active under daylight or UVA exposure. The addition of these anthraquinone derivatives to solutions and materials has been shown to produce significant amounts of ROS, resulting in light-induced antibacterial functions [7,8]. While successful in ROS generation, the possible leaching of these colorants from the materials needs to be addressed [8]. One means of addressing this issue is the covalent linkage of the light-active agents to the substrate. Chemical incorporation of light-active agents onto polymers has been explored as a successful method to prevent leaching. Surface modification and functionalization of cellulose with anthraquinone dyes have demonstrated durable light-induced antimicrobial properties on cotton fibers [11] while preventing the leaching of the agents from the fabrics. Applying this chemical modification to nanomaterials provides a means to expand the utility of these hybrid materials while preventing leaching-related complications.

In terms of potential carrier materials, cellulose nanocrystals (CNCs) stand out as prospective candidates, as they are produced from agricultural wastes and are considered renewable and sustainable materials [12,13]. Cellulose nanomaterials modified with a photoactive group also have promising biomedical applications. Cellulose nanocrystals (CNCs) are normally produced via the acid hydrolysis of cellulose, which results in nanometer-sized materials with unique structural features. These CNCs are functionalized with covalent sulfate ester groups on their surfaces due to the hydrolysis with sulfuric acid [14]. These sulfate groups make the CNCs easy to disperse in water but also hinder the incorporation of other functional groups [15,16]. Removal of the sulfate ester groups can be achieved following previously described procedures [17,18], and these partially desulfated CNCs can react with a light-active species, such as anthraquinone, to create non-leaching functionalized CNCs. 

This work reports the first success in directly and covalently linking an anthraquinone derivative onto the surfaces of CNCs. By utilizing a variety of analytical techniques, the chemical and morphological structures and characteristics of AQ-CNCs were investigated, and the results of their photoactive properties represent a significant step forward in the development of anti-microbial hybrid materials using CNCs. 

## 2. Materials and Methods 

### 2.1. Materials

Microcrystalline cellulose (Acros, Fair Lawn, NJ, USA), sulfuric acid (96.2%, Fisher Scientific, Fair Lawn, NJ, USA), sodium hydroxide (97%, Fisher Scientific), sodium chloride (99%, Fisher Scientific), anthraquinone-2-carboxylic acid (99%, Tokyo Chemical Industry CO., Tokyo, Japan), N,N’-carbonyldiimidazole (CDI) (98%, Oakwood Chemical, Estill, SC, USA), N,N-dimethylformamide (DMF) (99.9%, Sigma-Aldrich, Saint Louis, MO, USA), dimethyl sulfoxide (DMSO) (99.9%, Sigma-Aldrich), magnesium sulfate (97%, Sigma-Aldrich), potassium bromide (99%, Acros), poly-L-lysine solution (0.1 *w/v* in H_2_O, Sigma-Aldrich), and ethyl alcohol (95%, Sigma-Aldrich) were used as received without any further purification. Muscovite mica pieces were mechanically peeled to reveal fresh (0001) surfaces before use. AC240TS-R3 silicon cantilevers were purchased from Oxford Instruments (Santa Barbara, CA, USA). Asylum Research Water (≥18.2 MΩ) was purified using a Milli-Q system (Q-GARD 2, MilliporeSigma, Darmstadt, Germany).

### 2.2. Preparation of Cellulose Nanocrystals (CNCs)

Cellulose nanocrystals (CNCs) were prepared via sulfuric acid (H_2_SO_4_) hydrolysis of microcrystalline cellulose following a procedure reported previously [19]. The prevalence of sulfate groups on the surface was determined using conductometric titrations. The conductivity of the mixture was recorded using an Accumet XL600 benchtop meter (Fisher Scientific) with a conductivity electrode (AccuTupH conductivity/Temp probe, Fisher Scientific) against 0.02 M sodium hydroxide (NaOH). 

Removal of the sulfate ester groups was achieved as previously described with minor charges [17]. The CNC suspension with sulfate ester groups was hydrolyzed in concentrated NaOH for 5 h. The desulfated CNCs were then purified via repeated homogenization using DI water. Drops of 0.1 M H_2_SO_4_ were added into the mixture to neutralize the pH. The prevalence of residual sulfate groups on the desulfated cellulose nanocrystals (DS-CNCs) was determined using conductometric titration. Twenty-five milligrams of DS-CNCs were dispersed in 20 mL DI water and sonicated with an ultrasonic liquid processor S4000 (Misonix Inc., Farmingdale, NY, USA) for 5 min under 20% amplitude in an ice bath. The sulfated and desulfated cellulose nanocrystal suspensions were conditioned with 1 mM sodium chloride (NaCl) and then titrated with 0.02 M NaOH under continuous stirring. After each addition of NaOH, the conductivity was measured after 1 min to ensure that the conductivity value was stabilized. The conductivity of the mixture was recorded using an Accumet XL600 benchtop meter.

### 2.3. Surface Functionalization of CNCs with Anthraquinone-2-Carboxylic Acid

Synthesis of the anthraquinone-2-carboxylic acid (AQC)-modified CNCs was performed in two steps, as shown in Figure 1a. First, AQC was dispersed in 30 mL dried dimethylformamide (DMF). Then, 0.4004 g–0.8008 g of N,N’-carbonyldiimidazole (CDI) was dissolved in 10 mL of the dried DMF, and the solution was dropped into the AQC solution at 80 °C for 30 min under a N_2_ atmosphere to make the reaction mixture. Then, a total of 0.1 g DS-CNCs were dispersed in 10 mL dried DMF. After sonication at 30% amplitude for 5 min, the dispersed CNC suspension was added into the reaction mixture, followed by stirring at 80 °C for the specified reaction time under a N_2_ atmosphere. The final products were isolated via centrifugation and rinsed thoroughly with DMF and ethyl alcohol. Anthraquinone-2-carboxylic-acid-modified cellulose nanocrystal powder (AQ-CNC) was obtained after air-drying under dark conditions. 

### 2.4. Characterizations of the AQ Modified CNCs

Fourier transform infrared (FTIR) spectroscopy of cellulose nanocrystals before and after the functionalization was performed with a Nicolet 6700 FTIR spectrometer (Thermo Electron Co., Waltham, MA, USA) by using potassium bromide (KBr) pellets with a wavenumber range of 4000–400 cm^−1^, resolution of 4 cm^−1^, and 64 accumulations. The samples were mixed with KBr powder (2 mg sample mixed with 200 mg KBr) to prepare the pellets. The relative abundance of ester bonds on AQ-CNCs was calculated based on an intensity ratio of the ester bond (1725 cm^−1^) versus that of the C–H bond (2898 cm^−1^), as measured using FTIR spectroscopy [20]. 

The ratios (grafting yields) of the grafted anthraquinone moieties on the desulfated cellulose nanocrystals were quantified by measuring the absorbance of its DMSO suspension (0.200 g/L) at 331 nm with a UV-vis spectrophotometer Evolution 600 (Thermo Fisher Scientific, Waltham, MA, USA) based on a calibration curve of AQC with CNCs in DMSO solutions.

Products of the CNC functionalizations (CNC, DS-CNC, and AQ-CNC) were characterized using an atomic force microscope (MFP-3D, Asylum Research Corp., Santa Barbara, CA, USA). Silicon probes (AC 240-TS, Olympus America, Central Valley, PA, USA) with a nominal spring constant of 1.7 N/m, were used for imaging. Silicon probes were used after a brief cleaning in ethanol and drying under nitrogen as needed. All samples were imaged in tapping mode under ambient conditions. Poly-L-lysine-coated mica (0001) surfaces were prepared by first peeling the mica to expose the mica (0001) surface, then placing freshly cleaved mica in the 0.1% *w/v* poly-L-lysine solution for 10 min, and finally rinsing with MilliQ water [21]. CNC samples for the atomic force microscopy (AFM) imaging were prepared by dispersing the crystals in MilliQ water using an ultrasonic liquid processor S4000 (Misonix Inc.) for 2.5 min (5 s on and 5 s off) at 30% amplitude in an ice bath. Solutions of functionalized CNCs were prepared at 0.002% w/v. After dispersing the CNCs in each solution, a 40 µL solution was deposited onto the poly-L-lysine-coated mica (0001) surfaces and allowed to dry in a clean and light-shielded container for at least 12 h.

### 2.5. Detection of Hydroxyl Radicals 

The production of hydroxyl radicals by AQ-CNCs was evaluated in DI water under UVA exposure and without UVA exposure conditions. The yields of hydroxyl radicals for the varied synthesis conditions were indirectly quantified by measuring the photo-bleaching rates of a selective radical quencher, N,N-dimethyl-4-nitrosoaniline (p-NDA). The bleaching rate of p-NDA in each sample was determined by monitoring the absorption of the mixture at the wavelength of *λ*_max_ (440 nm) with a UV-vis spectrophotometer as a function of the illumination time. The UVA exposures of the mixtures were conducted in a Spectrolinker XL-1000 (Spectroline, Westbury, NY, USA), which is a UV cross-linker with five 8 W UVA lamps (365 nm wavelength). The distance between the light source and samples was 12 cm. The light intensity in the cross-linker was 3 mW/cm^2^.

### 2.6. Detection of Hydrogen Peroxide 

The hydrogen peroxide generated by AQ-CNCs in the aqueous system under UVA (365 nm) irradiation was also quantified using an iodometric method. Ten milliliters of 0.2 g/L AQ-CNC suspension samples prepared in a tube were irradiated under UVA light (365 nm wavelength) in the UV cross-linker with a light intensity of 3 mW/cm^2^. For every designated testing duration, 0.1 mL of the sample was taken from the tube and diluted to 1 mL with DI water. The diluted sample was mixed with 1 mL of reagent A (66 g/L potassium iodide (KI), 2 g/L sodium hydroxide (NaOH), and 0.2 g/L ammonium molybdate tetrahydrate ((NH_4_)_6_Mo_7_O_24_·4H_2_O)) and 1 mL reagent B (20 g/L potassium hydrogen phthalate (KHP)). The hydrogen peroxide reacted with iodide ions to give triiodide ions, which appeared as a dark brown color. The absorbance of the sample was measured spectrophotometrically at 351 nm with an Evolution 600 UV-vis spectrophotometer, and the amount of H_2_O_2_ was calculated according to a standard calibration curve. 

## 3. Results

### 3.1. Preparation and Characterization of the AQ-CNCs

In theory, cellulose nanocrystals (CNCs) should possess sufficient numbers of hydroxyl groups for the intended incorporation of anthraquinone. However, sulfate groups on the surfaces of CNCs introduced during the sulfuric-acid-catalyzed hydrolysis can potentially block the esterification reaction and reduce the yields of the reaction. Thus, a desulfation process by means of alkaline hydrolysis was applied to reduce or remove the sulfate groups on the CNCs. Conductometric titrations were performed to compare the prevalence of sulfate groups on CNCs before and after the desulfation process. The titration curves are shown in Figure 1b.

NaCl was added into the suspension of CNCs as an electrolyte and contributed to the initial conductivity of the solutions. The counter ions of the sulfate groups, namely, protons, contributed to the initial conductivity of the CNC samples and initially became reduced with the addition of sodium hydroxide. When the sodium cations substituted all the protons, the conductivity started to increase due to the further addition of sodium hydroxide. A subtle decrease and then immediate increase in conductivity was due to the reduced sulfate contents on the DS-CNCs, confirming the significant reduction of sulfate groups on the DS-CNCs. The surface charge density *σ* in mequiv/g can be used as an indirect indication of the prevalence of sulfate groups, which is calculated from the titration results using:(1)$$\sigma =\frac{c_{\mathrm{NaOH}}\times V_{\mathrm{NaOH}}}{c_{\mathrm{CNC}}\times \alpha_{\mathrm{CNC}}},$$
where $c_{\mathrm{NaOH}}$ is the concentration of the titrant, $V_{\mathrm{NaOH}}$ is the titrant volume at the equivalence point, $c_{\mathrm{CNC}}$ is the concentration of the CNC suspension, and $\alpha_{\mathrm{CNC}}$ is the amount of CNC suspension that was titrated [18]. The surface charge density decreased from 0.36 mequiv/g to 0.04 mequiv/g after the desulfation process, indicating a significant reduction in the prevalence of sulfate groups. The lower absorbance intensity of S=O bonds at 1238 cm^−1^ on the DS-CNCs is shown in the FTIR spectra in Figure 1c(II), which also confirmed the decrease in the number of sulfate groups after the desulfation process [22].

The DS-CNCs were reacted with anthraquinone-2-carboxylic acid (AQC) to produce AQ-grafted CNCs (AQ-CNCs) following an esterification reaction catalyzed by N,N’-carbonyldiimidazole (CDI), as shown in Figure 1a. The resulting AQ-CNCs were characterized using FTIR, and structural features of the ester and anthraquinone groups can be found on spectra (III) in Figure 1c. The peak at 1590 cm^−1^ was due to aromatic C=C stretching from the anthraquinone-2-carboxylic acid molecule and the peak of 1673 cm^−1^ was the carbonyl stretching peak on the anthraquinone group [11]. Another new peak at 1725 cm^−1^ was attributed to the ester group formed between DS-CNCs and AQC [11]. The band at 2898 cm^−1^ was the representative C–H bond in the cellulose, which was not affected by the reaction.

All materials were subject to high-resolution AFM imaging to visualize the fiber packing, as well as the morphology and structures. AFM was selected as the technology choice due to its intrinsic advantages: (a) true 3D information, (b) label-free, and (c) preserving the material’s native environment, in this case, ambient. AFM topographs are shown in Figure 2a–c, and examples of the individual CNCs are indicated in the images by oval enclosures. The AFM topographic images revealed that the individual crystals of each functionalization exhibited a rod-like geometry. The diameter and length of each rod highlighted in Figure 2a–c were measured from the topographic images and are summarized in Table 1 below. The diameters determined from the height of the AFM images of the CNCs, DS-CNCs, and AQ-CNCs fell within the range of 2.5–3.5 nm. Since individual CNCs exhibited very similar diameters and lengths upon treatments of desulfation followed by anthraquinone functionalization, the structural integrity of the individual fibers remained.

The AFM topographic images also revealed the presence of bundles of rods for each sample. Examples of these bundles are indicated by rectangular enclosures. In the two CNC bundles shown in Figure 2a, one was a cross of 131.5 and 110.5 nm rods, and the other was a short rod (77.4 nm) that was completely atop of another rod (173.8 nm), resulting in the bright contrast seen in the AFM topograph. In the two DS-CNC bundles shown in Figure 2b, the topmost bundle was composed of three rods: the bottom long rod (225.7 nm) that was completely under and supporting two short rods (51.7 and 70.5 nm, respectively), which appeared as two brighter contrasts. The top-left bundle was composed of three rods: two rods (219.1 and 169.1 nm) overlapped side-by-side, with the third rod (167.6 nm) stacking perpendicularly atop, leading to a cross shape. In the two AQ-CNC bundles, the bundle in the upper-right of Figure 2c consisted of a 68.1 nm rod completely stacked atop of a 117.4 nm rod. The bundle in the lower-left of Figure 2c consisted of three rods: a 91.9 nm long rod attached to the surface, with a small 56.7 nm rod aligned next to the right end, and the third 134.5 nm rod stacked at an angle on top of the end of the rod beneath it. In contrast to the CNCs and DS-CNCs, which exhibited branched bundles, the bundles in the AQ-CNC samples mostly consisted of aligned rods, which is advantageous for the subsequent production of composite materials. 

For all three types of CNCs, large aggregates were occasionally seen. In Figure 2d, the AFM amplitude image clearly revealed the individual CNC rods within the “flake-shaped” aggregate (maximum lateral dimension of 6.6 µm wide), which were aligned in parallel relative to each other. Simultaneously, the topographic image (not shown) revealed the height of the aggregate as 226.7 nm, which was equivalent to 86 layers of individual crystals. In Figure 2e, the AFM amplitude image showed the individual rods within the “haystack” aggregate of the DS-CNCs (with a maximum lateral dimension of 3.1 µm), adopting random orientations. The AFM topographic image (not shown) revealed that the height of the aggregate was 67.8 nm, which was equivalent to 21 layers of individual crystals. In Figure 2f, the AFM amplitude image revealed the “interwoven” arrangement of the individual AQ-CNC rods within the “tadpole-shaped” aggregate (maximum lateral dimension of 2.8 µm). The AFM topographic image (not shown) showed the aggregate’s height as 206.7 nm, which was equivalent to 75 layers of individual crystals. Even though the aggregates were a minority population, from this AFM investigation, we recommend caution to prevent them when processing AQ-CNCs, as the AQ-CNC aggregates would be more difficult to disperse due to the interwoven packing.

### 3.2. Effects of the Reaction Conditions on the Grafting Efficiency

In order to optimize the grafting efficiency of AQ onto the CNCs, the conditions for the two mains steps of the esterification reaction were varied. The first step of the esterification reaction was the formation of an AQC intermediate with the CDI, which was controlled by varying the molar ratio of CDI to AQC. For the second step of the reaction, the reaction time and ratio of anhydroglucose unit (AGU) to AQC and CDI were varied. To quantify the effects of these changes, the intensity ratios of the ester bond peak (1725 cm^−1^) to the C–H bond peak (2898 cm^−1^) were assumed to be equivalent to the relative abundance of covalently linked AQ groups on the CNCs’ surfaces. The ester peak was attributed to the covalent linkage of AQ to the CNCs and the C–H bond peak was unaffected by the reaction, and thus served as an internal standard [20]. In addition, UV-vis measurements of the anthraquinone groups on AQ-CNC samples were conducted and yields of the grafting reaction of AQC on the CNCs were calculated based on a calibration curve.

Figure 3a shows the FTIR band ratios of 1725 cm^−1^/2898 cm^−1^ and grafting yields under varied molar ratios of AQC/CDI. With the increase of the CDI amount in the reaction system, the peak ratios of 1725 cm^−1^/2898 cm^−1^ in the FTIR spectra of the produced AQ-CNC samples increased correspondingly. The CDI reacted with the carboxylic acid group in the AQC and was highly sensitive to moisture. By increasing the amount of CDI relative to AQC, the losses due to moisture during the reaction could be compensated for. Measurements of UV-vis spectrometry of the AQ-CNC samples also showed an increase of the grafting yields of AQC on the CNCs (Figure 3a) with increasing CDI, verifying its importance to the incorporation of AQC onto the CNC surfaces. 

To further optimize the reaction yield, the AGU/AQC/CDI molar ratio to the grafting esterification reaction was investigated. However, when the AGU/AQC ratio was changed, the amount of CDI also needed to be changed such that the AQC/CDI relationship was kept consistent to enable the maximum formation of the AQC intermediate. To this end, the AQC/CDI ratio was kept at around 1:2–1:2.5 when the AGU/AQC ratio was changed from 1:1 to 1:4. Such a small variation in the ratio of AQC/CDI should have little impact on the yield changes seen in the varied ratios of AGU/AQC. The FTIR results indicated that the esterification reaction was improved as the AGU/AQC ratio was increased in the system, as evidenced by the increased esterification ratios and grafting yields shown by the peak ratios of 1725 cm^−1^/2898 cm^−1^ (Figure 3b), which was verified by the UV-vis absorbance of AQ-CNCs. Higher concentrations of AQC means that it was more readily available for incorporation and helped to drive the reaction forward, resulting in an increase in reaction yields. Thus, both ratios of AQC/CDI and AQC/AGU were important for optimizing the esterification reaction and driving the reaction forward to increase product formation.

The esterification products (AQ-CNCs) also increased as the reaction time increased, as shown in Figure 3c. The molar ratio of AGU/AQC/CDI was kept constant at 1:4:8, yet the ratio of 1725 cm^−1^/2898 cm^−1^ increased from 0.380 within 2 h to 0.777 at 24 h, doubling the yield. The grafting ratios characterized by the UV-vis spectroscopy verified the same trend of increasing AQ incorporation as the reaction time increased from 2 to 24 h. Both measurements of AQ incorporation supported the conclusion that increased exposure of AGU to AQC led to increased incorporation of AQ on the CNCs. 

### 3.3. Photoactivities of AQ-CNCs

Anthraquinone derivatives possess photoactivities, and when AQ was chemically incorporated onto cellulosic materials, the desired light-induced antibacterial functionality was achieved. The overall photo-initiated functional process was caused by the excitation of AQ to its excited singlet state, which quickly underwent intersystem crossing to its triplet state. The triplet AQ abstracted a hydrogen atom from a weak C–H bond in the cellulose to become AQH radical, which reacted with triplet oxygen to result in reactive oxygen species, such as hydroxyl radicals and hydrogen peroxide [11]. To verify that this process remained intact and that the AQ-CNCs possessed the light-induced functionality, the amount of reactive oxygen species (ROS) produced needed to be evaluated. One of the ROS produced, hydroxyl radicals, could be detected and quantified using an indirect spectrophotometrical method involving a hydroxyl radical scavenger, p-nitrosodimethylaniline (p-NDA) [7]. The bleaching rates of p-NDA for the varied synthesis conditions of AQ-CNCs were determined by monitoring the absorption of each solution at a wavelength of 440 nm after exposure to UVA (365 nm) light. The results are shown in Figure 4a,b.

According to the results shown in Figure 4a, the photo-bleaching rate accelerated as the ratio of AGU/AQC/CDI increased, which was consistent with the photoactivity of the AQ structure and increased AQ loading on the AQ-CNCs. When the UVA irradiation time increased, the amount of hydroxyl radicals cumulatively increased as well. However, in Figure 4b, after a reaction time of 24 h, the AQ-CNC sample made with an AGU/AQC/CDI molar ratio of 1:4:8 with a measured grafting ratio of 7.454 wt.% demonstrated a similar but slightly lower photo-bleaching power than that of the sample made in a molar ratio of 1:4:6 with a grafting ratio of 4.428 wt.%. This phenomenon revealed a potential saturation of the photoactivity on the AQ-CNCs over time. Too high of a concentration of AQ groups on AQ-CNCs may interfere with the formation of the triplet structure of AQ and inhibit the abstraction of hydrogen atoms from the connected cellulose rings since all AQ structures are chemically grafted in close vicinity of the surfaces of the CNCs.

To further verify the photoactivity of AQ-CNCs, the hydrogen peroxide produced in an aqueous condition was measured by using an iodometric method under UVA light irradiation. The absorbance of the mixture was measured spectrophotometrically at 351 nm, and the amount of H_2_O_2_ in the DI water was calculated according to a standard calibration curve. The results are shown in Figure 5a,b.

Based on Figure 5a, the amounts of hydrogen peroxide formed increased as the UVA exposure time was prolonged. The AQ-CNC samples containing more AQC units generated more hydrogen peroxide in the solutions, which was consistent with the p-NDA bleaching results. However, in Figure 5b, the AQ-CNC sample with a grafting rate of 4.428 wt.% exhibited similar amounts of hydrogen peroxide in comparison to that of the one with the highest grafting ratio of 7.454 wt.% at 90 min of exposure time. This outcome matched the previous abnormal results of the measured hydroxyl radicals, indicating the saturation of photoactivities at high concentrations of AQ loading at higher exposure times. Further investigation of the photoactivity of AQ-CNCs revealed an interesting phenomenon from these materials, i.e., the continued generation of hydroxyl radicals even after the UVA irradiation was terminated. The three AQ-CNC samples containing 2.333 wt.% (1:4:4 molar ratio), 4.428 wt.%, and 7.454 wt.% of AQ moieties from Figure 5b were tested, and all exhibited persistent photo-bleaching ability against p-NDA after the UVA exposure was terminated (Figure 6b), indicating the generation of hydroxyl radicals without direct light irradiation. The more AQ units grafted onto the CNCs, the better the photo-bleaching efficiency of the samples was, indicating that a sustained ROS generation efficacy was dependent on the AQ loading on the CNCs. This unexpected phenomenon could be caused by an unusual property possessed by some anthraquinone derivative structures, the so-called light-absorbing transient behavior [23,24].

A proposed mechanism for this light-absorbing transient behavior is shown below in Figure 6a. The light-absorbing transient (LATs) are quasi-stable intermediates resulting from relatively stable radical terminations [23,24]. The excited triplet AQ structures on the AQ-CNCs could abstract a hydrogen atom from a hydrogen donor site on CNCs to form AQH radical intermediates (Figure 6a), which are the light-absorbing transient structures similar to that of benzophenone radicals reported in References [23,24]. Due to the delocalization of electrons in the AQH radicals (as explained by the existence of resonance structures), these LAT structures are more stable. The light-absorbing transient structures can react with oxygen and other species, generating reactive oxygen species (ROS), such as hydroxyl radicals, perhydroxyl radicals, and hydrogen peroxide [7,11]. The oxygen consumption plays an important role in the photoreaction in the presence of hydrogen donor. It has been reported that the decay of the light-absorbing transient photoproducts is accelerated with exposure to oxygen [25,26].

### 3.4. Stability of Light-Active AQ-CNCs

Based on the photo-bleaching reactions of p-NDA demonstrated by the AQ-CNC samples under or without UVA exposure, the formation of excited AQ and radical AQH (and conjugate) structures in ambient conditions was key. The CNCs served as effective hydrogen donors (R–H) directly connected to AQ, enabling easier hydrogen abstraction and thus the formation of AQH. The ease of AQH formation enabled by the CNCs’ proximity may help to compensate for the radical AQH quenching caused by the oxygen exposure, leading to prolonged lifetimes of the radical AQH and the persistent generation of ROS under darkness. Interestingly, the AQH transient structures were quite stable and could store the light-induced energy for several days, as most of the tests on the AQ-CNCs in the dark were conducted several days after being sealed in darkness.

## 4. Conclusions

We have successfully demonstrated a direct esterification reaction between cellulose nanocrystals (CNCs) and anthraquinone carboxylic acid (AQC) for the first time. This covalent binding was verified using FTIR and UV-vis spectroscopy. The morphology and photoactivity were characterized following the covalent reactions. AFM revealed that the CNCs retained their structural integrity, which is an advantage for the subsequent production of composite materials. The AQ-CNCs were highly photoactive under daylight or UVA, generating both hydroxyl and hydrogen peroxide radicals. The amount of radicals increased with increasing UVA exposure time or concentration of AQC on the CNCs up to a point. More interestingly, AQ-CNCs formed light-absorbing transient structures that stored the light-excited structures and provided effective bleaching of p-NDA under darkness. The AQ-CNC-based materials offer a promising candidate for developing green, photoactive hybrid materials with self-disinfecting properties.

## Figures and Tables

**Figure 1 materials-13-03547-f001:**
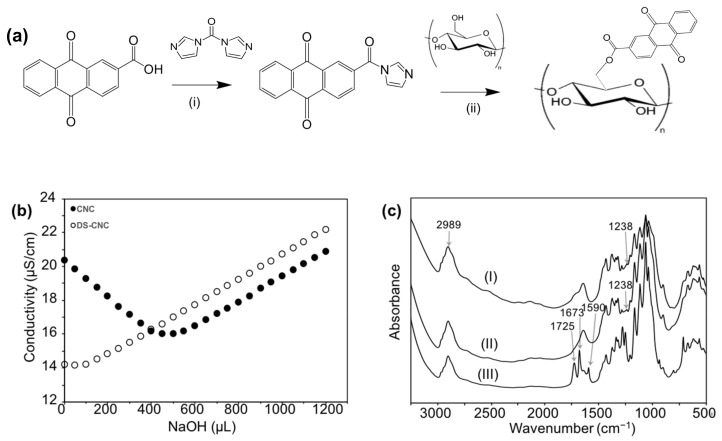
Preparation and characterization of the desulfated CNCs (DS-CNC) and AQ-CNCs. (**a**) Synthesis of AQ-CNCs: (i) DMF at 80 °C for 0.5 h and (ii) DMF at 80 °C for 2, 8, 12, and 24 h. (**b**) Conductometric titration curves of the CNC samples. (**c**) FTIR of the (I) CNCs, (II) DS-CNCs, and (III) AQ-CNCs.

**Figure 2 materials-13-03547-f002:**
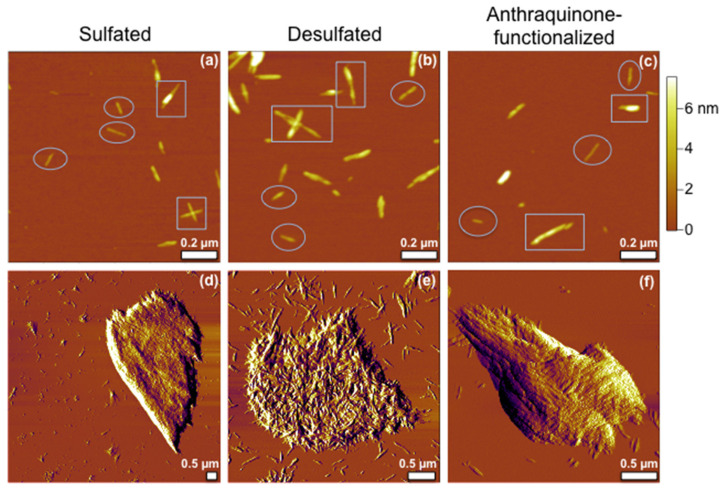
AFM topographic images (1.2 µm × 1.2 µm) of (**a**) CNCs, (**b**) DS-CNCs, and (**c**) AQ-CNCs. All images were acquired using tapping mode in ambient conditions. Scale bars: 0.2 µm. AFM amplitude images of aggregates are shown for (**d**) CNC, (**e**) DS-CNC, and (**f**) AQ-CNC samples. Scale bars: 0.5 µm.

**Figure 3 materials-13-03547-f003:**
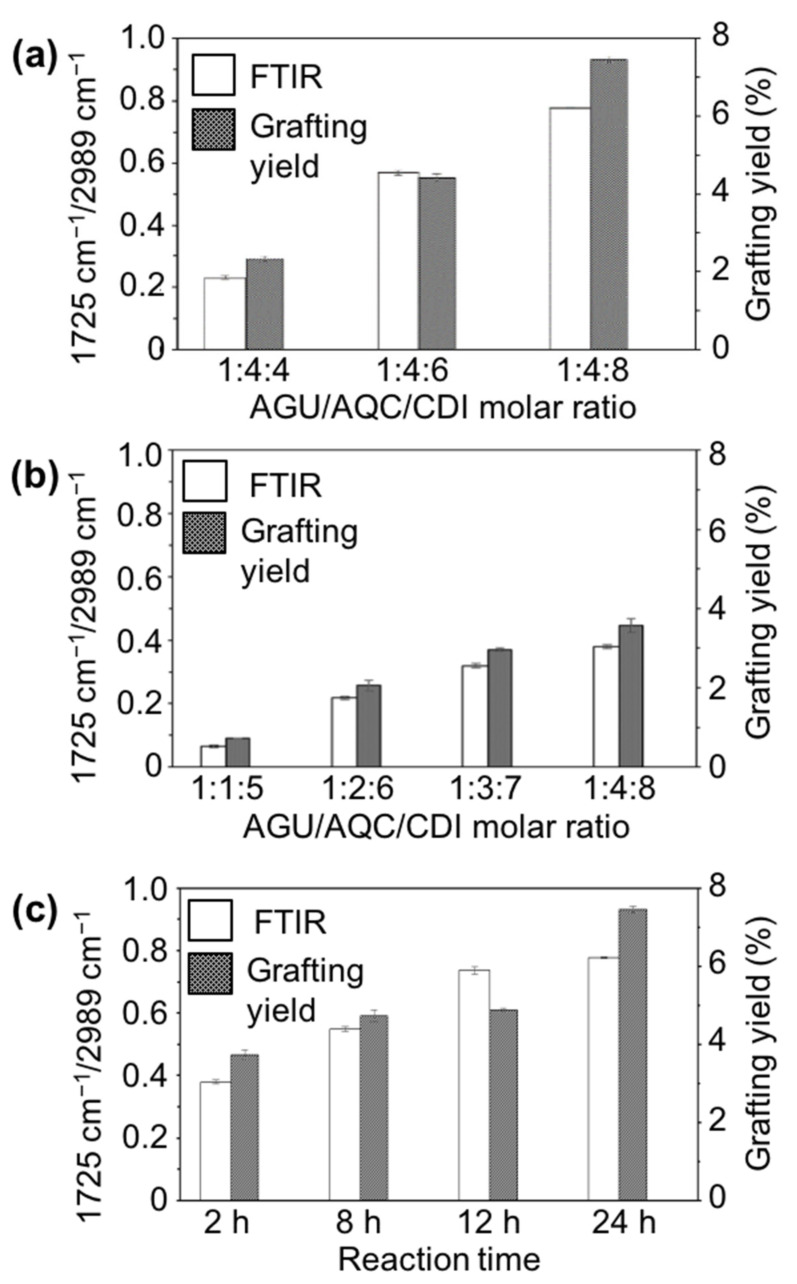
Impacts of reaction conditions on ester formation, as shown in the ratios of 1725 cm^−1^/2989 cm^−1^ and grafting yields: (**a**) impact of ratios of AQC/CDI (AGU/AQC unchanged), (**b**) impact of ratios of AGU/AQC/CDI, and (**c**) impact of reaction time.

**Figure 4 materials-13-03547-f004:**
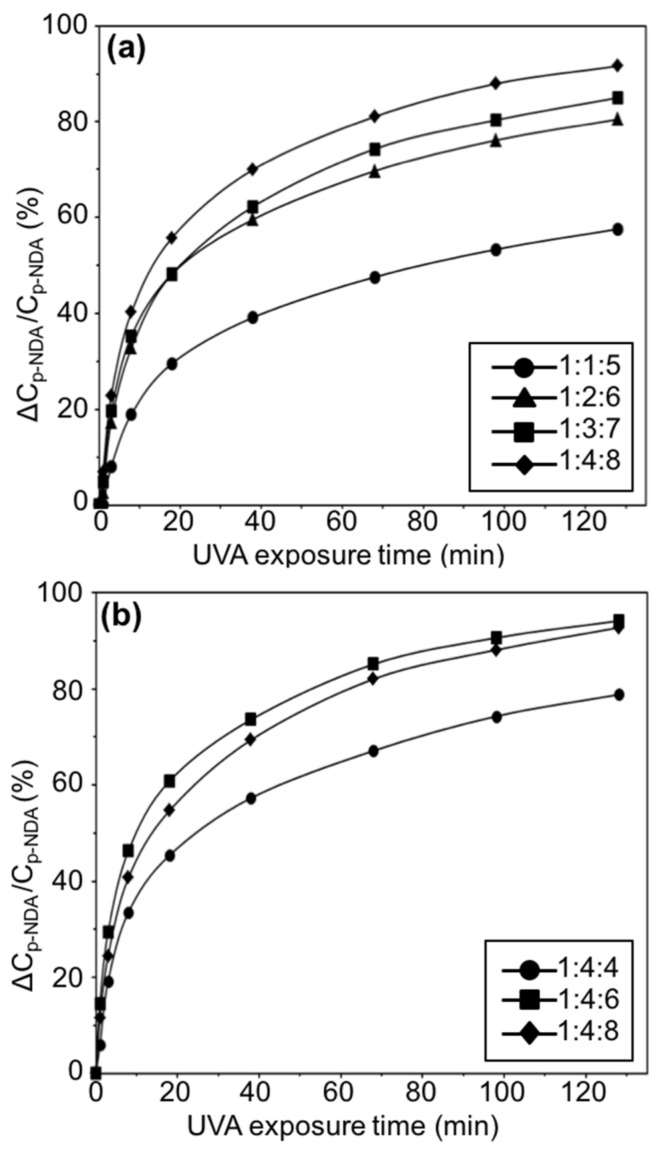
Photo-bleaching rates of p-NDA under UVA by AQ-CNCs made from (**a**) varied AQC/AGU molar ratios at a reaction time of 2 h and (**b**) varied AQC/CDI molar ratios at a reaction time of 24 h.

**Figure 5 materials-13-03547-f005:**
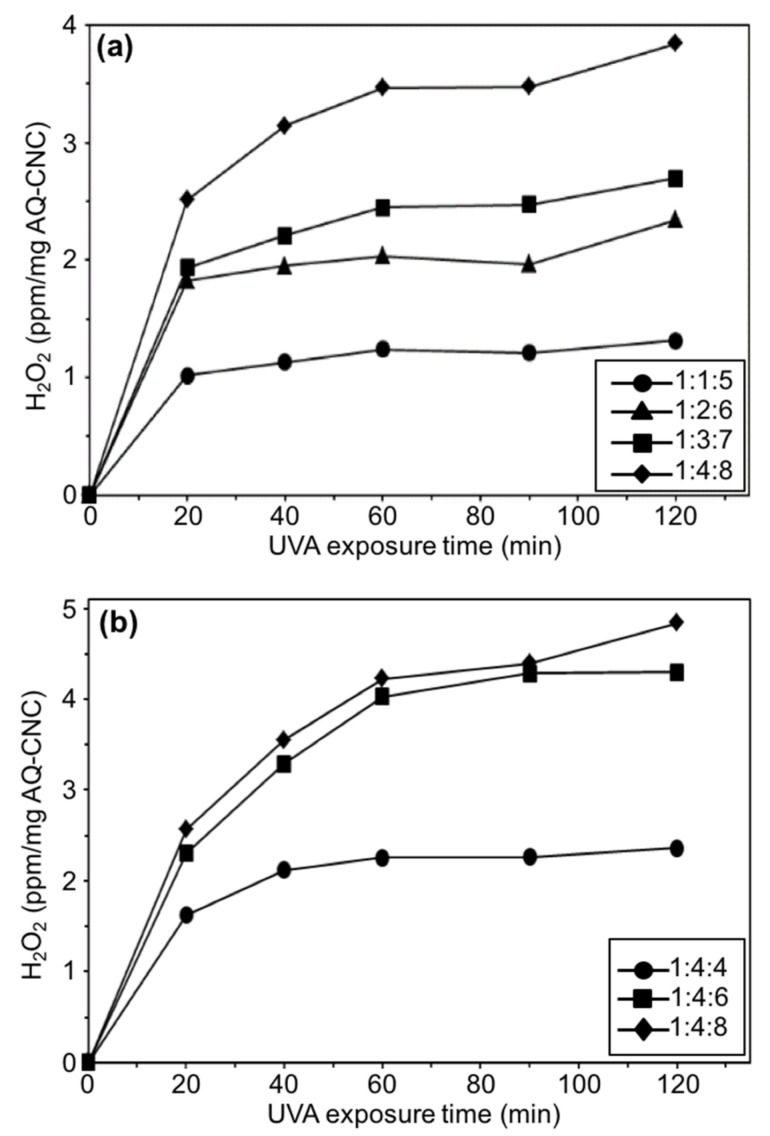
Amounts of hydrogen peroxide produced under UVA (**a**) by AQ-CNCs made from varied AGU/AQC molar ratios at a reaction time of 2 h and (**b**) by AQ-CNCs made from varied AQC/CDI molar ratios at a reaction time of 24 h.

**Figure 6 materials-13-03547-f006:**
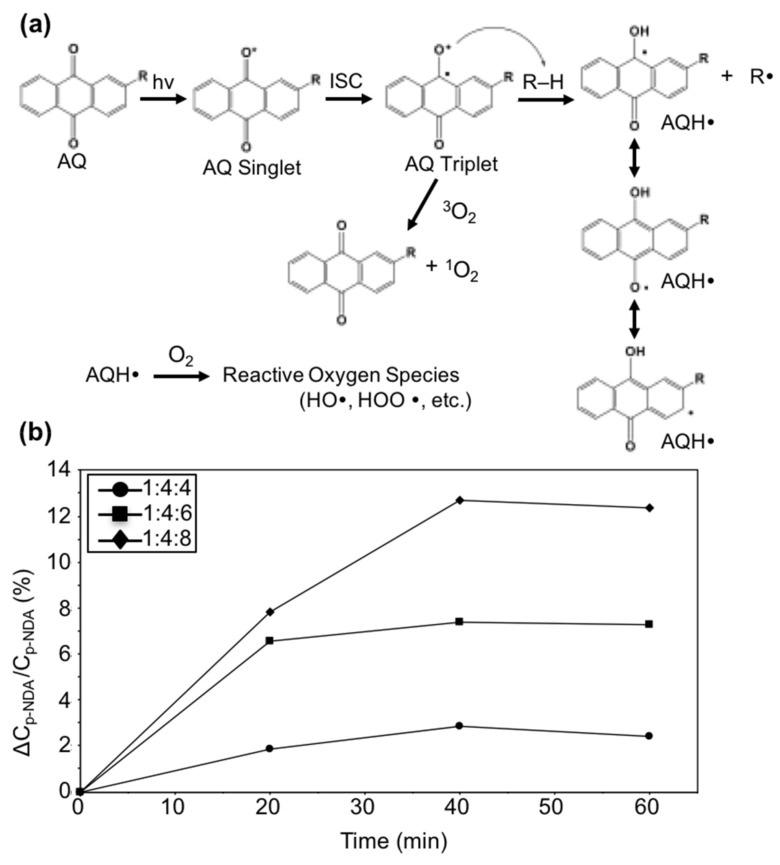
(**a**) Scheme of the photo-reaction of AQ derivatives with R–H (cellulose) and the formation of ROS. (**b**) Generation of hydroxyl radicals in darkness by AQ-CNCs from varied AQC/CDI molar ratios.

**Table 1 materials-13-03547-t001:** Dimensions of individual CNCs measured from Figure 2a–c.

Samples	Diameter (nm)/Length (nm)		
CNC	2.8/115.6	2.5/80.6	2.6/72.9
DS-CNC	3.0/89.7	3.5/73.4	3.1/118.6
AQ-CNC	3.0/99.1	2.8/61.6	2.5/132.1

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
