# Peer review of "Daylight-Active Cellulose Nanocrystals Containing Anthraquinone Structures"

_materials, 2020, doi:10.3390/ma13163547_

Round 1

Reviewer 1 Report

The manuscript deals with antimicrobial and anti-viral materials in particular using anthraquinone and cellulose nanocrystals as the photoactive and carrier components. The morphology and the photoactive properties have been investigated and an unexpected generation of reactive oxygen species have been evidenced.

The manuscript is well organized and clear.

Just two points should be improved:

INTRODUCTION. Please reorder reference starting from [1], not [6]… explaining the reason of each citation. Multiple citations are allowed but not so extensive, for example, [1-6], [5-9] and so on.

REFERENCES. They are not in compliance with the journal. See “Instruction for authors”. Furthermore in my opinion the bibliography is not up to date. Many cited articles are aged (1980, 1993, 1996, 1998 and so on).

Author Response

Reviewer: 1

The manuscript deals with antimicrobial and anti-viral materials in particular using anthraquinone and cellulose nanocrystals as the photoactive and carrier components. The morphology and the photoactive properties have been investigated and an unexpected generation of reactive oxygen species have been evidenced.

The manuscript is well organized and clear.

We thank the reviewer for their kind remarks.

Just two points should be improved:

Point 1: INTRODUCTION. Please reorder reference starting from [1], not [6]… explaining the reason of each citation. Multiple citations are allowed but not so extensive, for example, [1-6], [5-9] and so on.

We took the reviewer’s suggestion, made certain that references now appear sequentially and correspondence with text. We have also removed some of the citations in the [1-6] and [5-9] group so that the citations are not too extensive. We have highlighted any changed or substituted references in the manuscript for editorial clarity. The explanations of each citation are listed below.

[1-3]: [1]: Foorginezhad, S.’s group prepared self-cleaning and durable super-hydrophobic fabric by spraying vinyl-modified TiO2 sol and PDMS solution. [2]: Doganli, G.’s team used titanium dioxide (TiO2) as coating compound to add self-cleaning and antibacterial properties to the cotton fabric. [3]: Ma. Y’s team reported an approach, chemically incorporating both 3,3′,4,4′-benzophenonetetracarboxylic dianhydride (BPTCD) and [2-(methacryloyloxy)ethyl]dimethyl-(3-sulfopropyl)ammonium hydroxide (SBMA) onto Poly(vinyl alcohol-co-ethylene) (EVOH) nanofibrous membranes (NFMs). The resulted membranes could reduce microbial contamination and biofilm formation.

[4]: In Zhuo, J.’s dissertation, she prepared light-active functional polymer materials with incorporation of light reactive groups and successfully demonstrated the light-induced antimicrobial functions against both E. coli (gramnegative) and S. aureus (gram-positive).

[5,6]: Gao, A.’s team and Zhuo, J. studied the mechanisms of reactive oxygen species generation excited by long wavelength ultraviolet light or daylight on many photoactive chemicals

[7-10]: [7]: Liu, N. explained the mechanism that only light, oxygen and hydrogen donor are required for reactive oxygen species generation with the presence of photoactive anthraquinone compounds. [8]: Zhuo, J. verified the light-induced biocidal function of the modified cotton fabrics under the UVA light (365 nm) illumination. [9]: Gorner, H further explored the photolysis of 9,10-anthraquinone (AQ), 2-methyl- and 2,3-dimethyl-AQ in air-saturated acetonitrile-water in the presence of various donors: formate, ascorbic acid, alcohols. [10]: Zhang, Z. and his team discovered the photoactive vitamin K (VK) compounds which could generate reactive oxygen species of hydroxyl radicals and singlet oxygen under daylight, UVA, and UVB irradiations. The possible photoreaction paths could exist in an environment of good or poor hydrogen donors.

[7,8]: [7]: Liu, N. found that 2-Anthraquinone sulfonate and two other similar compounds produced singlet oxygen, hydroxyl radicals, and superoxide, so-called reactive oxygen species (ROS), under UVA exposure. [8]: Zhuo, J. incorporated two anthraquinone derivatives onto cotton fabrics by a vat dyeing process and demonstrated light-induced biocidal functions.

[8]: Zhuo, J. dyed fabrics with two light-active chemicals: 2-ethylanthraquinone (2EtAQ) and Vat Yellow GCN, she demonstrated light-induced biocidal functions however the colorfastness was concerning.

[11]: Liu, N. and her team developed self-cleaning cotton fabrics via chemically incorporating photosensitive 2-anthraquinone carboxylic acid (2-AQC) onto the fibers through a mild and efficient esterification reaction and they demonstrated excellent photo-induced self-cleaning properties.

[12,13]: [12]:The article explores the importance of cellulose nanocrystals as agriculture waste and the pre-treatments, methods involved in the production, the properties of cellulose nanocrystals prepared from crop and industrial wastes. [13] This research enriches the information on thermal stability and nanomechanical performance of cellulose nanomaterials, and provides increased knowledge on understanding the effect of cellulose nanocrystals as a matrix or reinforce in composites.

[14]: In Jonoobi, M.’s review, different cellulose nanocrystal preparation methods were covered. The sulfuric acid hydrolysis method introduces a high number of negatively charged sulfate groups on the surface of CNCs.

[15,16]: [15]: Natterodt, J. C. mentioned the high surface energy of CNCs is a result of the hydroxyl and sulfate half ester groups. They need some pretreatment steps for further functionalization. [16]: Börjesson, M. pointed out that cellulose nanocrystals (CNCs) prepared via sulfuric acid hydrolysis are always decorated with sulfate groups that yield a stable water suspension.

[17,18]: [17]: Hasani, M.’s team removed the sulfate ester groups completely by concentrated NaOH hydrolysis prior to further functionalization. [18]: The solvolytic desulfation of H2SO4-hydrolyzed CNCs was carried out by Jiang, F..

Point 2: REFERENCES. They are not in compliance with the journal. See “Instruction for authors”. Furthermore in my opinion the bibliography is not up to date. Many cited articles are aged (1980, 1993, 1996, 1998 and so on).

We have updated the citation style to follow the MDPI style in compliance with the journal. Additionally, we have removed all but two of the citations that are dated before 2000 and replaced them with publications less than 5 years old where possible.

Reviewer 2 Report

This paper deals with the development of cellulose nanocrystals containing anthraquinone structures for radical generation under illumination. The paper is well-written, clear and concise. There are a nice set of characterizations. It fits the scope of Materials. I suggest publication after only minor corrections:

  • Figures 1 are fuzzy, especially (b) and (c).
  • Still in Figure 1c, the x-axis is wavenumber not wavenumbers.
  • A comparison with the literature concerning the radical generation would be nice.

Author Response

Reviewer 2:

This paper deals with the development of cellulose nanocrystals containing anthraquinone structures for radical generation under illumination. The paper is well-written, clear and concise. There are a nice set of characterizations. It fits the scope of Materials.

We thank the reviewer for their kind remarks.

I suggest publication after only minor corrections:

Point 1: Figures 1 are fuzzy, especially (b) and (c). Still in Figure 1c, the x-axis is wavenumber not wavenumbers.

Figures 1(b) and 1(c) have been updated to a higher resolution version. The typo on the x-axis in Figure 1(c) is now labelled as “wavenumber”.

Point 2: A comparison with the literature concerning the radical generation would be nice.

We have added comparison, as suggestion in the beginning of the paragraph on page 11.

Reviewer 3 Report

The manuscript by Sun and co-workers discusses the preparation of anthraquinone grafted crystalline cellulose. The manuscript is well written and the results are well presented.

Detailed and accurate discussions are given on all results. All the claims and conclusions of the work are supported by evidence. The structure and editing of the manuscript is also up to the publication standards.

The topic itself fits well within the scope of the journal. The main question addressed by the research is antibacterial CNC, which is a timely and interesting research topic. The research topic is emerging and CNC is becoming popular for many applications.

The approach of cellulose modification is facile and new compared with other published approaches. The paper is well written and constructed. The experiments were carefully designed and well-explained.

Author Response

Reviewer 3:

The manuscript by Sun and co-workers discusses the preparation of anthraquinone grafted crystalline cellulose. The manuscript is well written and the results are well presented.

Detailed and accurate discussions are given on all results. All the claims and conclusions of the work are supported by evidence. The structure and editing of the manuscript is also up to the publication standards.

The topic itself fits well within the scope of the journal. The main question addressed by the research is antibacterial CNC, which is a timely and interesting research topic. The research topic is emerging and CNC is becoming popular for many applications.

The approach of cellulose modification is facile and new compared with other published approaches. The paper is well written and constructed. The experiments were carefully designed and well-explained.

We thank the reviewer for their compliment to our work and writing.

Reviewer 4 Report

This manuscript describes daylight-active cellulose nanocrystals Containing Anthraquinone structures which  and this findings could be helpful in developing green, photoactive hybrid materials with self-disinfecting properties. This manuscript is well-written, with sufficient reference and systematic data and relevant discussion, which can provide very useful information for future research. This article is highly recommended to be published on Journal of Materials. The following suggestions could be addressed in the revised version.

  1. The introduction part can be modified with recent up to date citations. Most of the citations are more than five years old.
  2. The AFM images were included. Authors may include SEM images for better visualization of the morphologies.

Author Response

Reviewer 4:

This manuscript describes daylight-active cellulose nanocrystals Containing Anthraquinone structures which and this findings could be helpful in developing green, photoactive hybrid materials with self-disinfecting properties. This manuscript is well-written, with sufficient reference and systematic data and relevant discussion, which can provide very useful information for future research. This article is highly recommended to be published on Journal of Materials.

We thank the reviewer for their recognition of this work.

The following suggestions could be addressed in the revised version.

Point 1: The introduction part can be modified with recent up to date citations. Most of the citations are more than five years old.

We have removed all but 2 of the citations dated before 2000 and replaced them with publications less than 5 years old where possible. The remaining citations, that are over 5 years old, are directly relevant to our work and provide essential background for readers to understand our scientific messages, thus kept as is.

Point 2: The AFM images were included. Authors may include SEM images for better visualization of the morphologies.

We understand that SEM has been a frequently used method, and we do indeed have SEM among all characterization tools. For this work, AFM was chosen over SEM for morphological characterization due to its intrinsic advantages: (a) AFM provides true 3D and high-resolution images, while SEM only provides 2D information; (b) AFM is label free, and probes non-conductive materials under their native conditions (e.g. ambient or liquid). In contrast, SEM often requires metal coating, and almost always requires high vacuum, which raise questions on the structural integrity. The justification is now added in the beginning of the last paragraph on page 5. We hope the reviewer will be convinced that AFM is the best choice to characterize the natural CNC morphology.